# A Deep Learning Model for Evaluating Meibomian Glands Morphology from Meibography

**DOI:** 10.3390/jcm12031053

**Published:** 2023-01-29

**Authors:** Yuexin Wang, Faqiang Shi, Shanshan Wei, Xuemin Li

**Affiliations:** 1Beijing Key Laboratory of Restoration of Damaged Ocular Nerve, Department of Ophthalmology, Peking University Third Hospital, Beijing 100191, China; 2State Key Laboratory of Virtual Reality Technology and Systems, School of Computer Science and Engineering, Beihang University, Beijing 100191, China; 3Beijing Tongren Eye Center, Beijing Tongren Hospital, Beijing 100051, China

**Keywords:** deep learning, meibomian gland, meibography, dry eye

## Abstract

To develop a deep learning model for automatically segmenting tarsus and meibomian gland areas on meibography, we included 1087 meibography images from dry eye patients. The contour of the tarsus and each meibomian gland was labeled manually by human experts. The dataset was divided into training, validation, and test sets. We built a convolutional neural network-based U-net and trained the model to segment the tarsus and meibomian gland area. Accuracy, sensitivity, specificity, and receiver operating characteristic curve (ROC) were calculated to evaluate the model. The area under the curve (AUC) values for models segmenting the tarsus and meibomian gland area were 0.985 and 0.938, respectively. The deep learning model achieved a sensitivity and specificity of 0.975 and 0.99, respectively, with an accuracy of 0.985 for segmenting the tarsus area. For meibomian gland area segmentation, the model obtained a high specificity of 0.96, with high accuracy of 0.937 and a moderate sensitivity of 0.751. The present research trained a deep learning model to automatically segment tarsus and the meibomian gland area from infrared meibography, and the model demonstrated outstanding accuracy in segmentation. With further improvement, the model could potentially be applied to assess the meibomian gland that facilitates dry eye evaluation in various clinical and research scenarios.

## 1. Introduction

Dry eye disease (DED) is a prevalent and multifactorial ocular surface disease that causes various ocular symptoms and visual impairment [1]. It is characterized by disrupted tear film homeostasis, and the tear film instability is regarded as a significant entry point to the pathogenesis of DED vicious circle [2,3]. The tear film instability is frequently caused by a lack of lipids or abnormal quality of lipids secreted by meibomian glands [4]. Meibomian glands are located in the tarsal plates of the upper and lower eyelids with a tubulo-acinar structure, and their dysfunction commonly manifests as gland atrophy and loss [5]. Thus, the evaluation of meibomian gland morphology is essential in DED assessment.

Infrared meibography is commonly applied to assess the morphological structure of the meibomian gland [6]. Generally, doctors manually grade the meibography according to the degree of meibomian gland loss. There are several commonly applied grading scales, including 0–3 grades and 0–4 grades [7,8]. However, manual grading has intra- and inter-grader disparity that impacts the repeatability and reliability of the assessment. Additionally, the discontinuous grading system cannot evaluate the meibomian gland loss with more accuracy. The unmet demand could be solved by developing an objective, accurate, automatic meibomian gland segmentation system.

In recent years, artificial intelligence has been wildly applied in clinical diagnosis, examination recognition, and prognosis prediction [9]. Several studies have applied traditional artificial intelligence methods including edge detection and contour extraction, for meibomian gland segmentation and area calculation, and the algorithm achieved outstanding consistency with the manual label [10,11,12]. However, these methods require complicated image preprocessing before the recognition and identification is affected by the image quality. Compared with the traditional method, deep learning (DL) models could learn the features automatically without manually defining them and have been shown to achieve excellent performance in various ophthalmological fields [13,14,15]. Limited studies have attempted to apply a convolutional neural network to grade meibomian gland loss automatically [16,17]. However, these studies considered the inter-gland district as the meibomian gland area, so they might underestimate the loss area, and training lacks images with severe meibomian gland loss that would affect the generalization of the model. In Md et al.’s study, it is also proposed that an automatic meibomian gland segmentation system be based on deep learning. However, we applied a pre-trained VAE-GAN as a quality inspector, embedding the shape constraint information into the neural network to improve accuracy [18].

The present study further included infrared meibography and annotated the contour of each meibomian gland and tarsus area. Then we trained a convolutional neural network-based deep learning model to automatically identify the meibomian gland and tarsus area and compare the results with the manual label. The model could be potentially applied to assess the meibomian gland that facilitates dry eye evaluation in various scenarios.

## 2. Materials and Methods

### 2.1. Dataset

The study performed adhered to the tenets of the Declaration of Helsinki, and the research protocol was approved by the Ethics Committee of Peking University Third Hospital. Informed consent was exempted because it was a retrospective study. In order to protect patients’ privacy, the personal information in meibography images was deleted.

We included 1434 meibography images from 366 patients diagnosed with DED in the Department of Ophthalmology of Peking University Third Hospital according to Tear Film and Ocular Surface Society Dry Eye Workshop II [2]. The meibography was captured in a standardized method with the Keratograph 5M (Oculus, Wetzlar, Germany), a non-contact corneal topographer based on the Placido ring and 880 nm infrared illumination. The surrounding light conditions were the same. The patients placed their chin on the chin rest, and the examiner everted the upper or lower eyelids to expose the palpebral conjunctiva adequately. The capture was focused on the meibomian gland and included the entire upper or lower palpebral conjunctiva in the middle of the meibography. The examination was performed by the same examiner. A preliminary review of the enrolled meibography was conducted, and the images were excluded if they (1) did not properly focus on the meibomian gland; (2) presented apparent finger images; or (3) captured the meibomian gland without sufficiently everting the eyelid. Following the initial review, 1087 meibographies were included in the analysis with 629 and 458 upper and lower eyelid images, respectively.

### 2.2. Data Labeling

The ground truth was the label manually drawn by human experts. We developed an online annotation software based on Microsoft COCO online platform with Python and JavaScript. The software could support user management, labeling, data review, and output. Thus, users could perform the annotation and conveniently review the label online with their laptops. The software has a lasso function that could define the contour for the area of interest. The human expert labeled the border of the tarsus area and the meibomian gland. For the upper tarsus, the upper border was set as the posterior lid margin, and the lower border was the edge of the proximal tarsal plate. For the lower tarsus, the upper border was defined at the fornix of the everted eyelid, and the lower border was also at the posterior lid margin. The horizontal borders were the intersection between the top and bottom borders. For the meibomian gland labeling, the border of each gland was drawn. Figure 1 shows the representatives of a meibograph and its annotation.

### 2.3. Network Development and Training

The meibography images were randomly allocated into three datasets, the training set (70%), validation set (20%), and test set (10%). One patient could only appear in one set. The training set was applied to train the network to fit the label better. The validation set facilitated tuning the model hyperparameters. Finally, the test set was used to evaluate the performance of the model. The initial image was cropped to 420 × 890 pixels, excluding the personal and device information. Before the training, we applied data augmentation to expand the training dataset, including color jittering, random lighting, horizontal flip, and Gaussian balance. After data augmentation, the data volume was expanded to 16 times that of the original data. Smoothing filtering with a bilateral filter with a kernel size of 3 and max-min value normalization was performed for pre-processing.

The framework of the deep learning model is shown in Figure 2. The model mainly consists of three parts, the upper and lower eyelid classifier, tarsus area segmentation, and meibomian gland segmentation. We first trained a ResNet18 model to classify the image as the upper or lower eyelid to achieve a more accurate segmentation based on the morphological characteristics of the upper and lower eyelids. The ResNet18 model leveraged 18 convolution blocks and the last binary classifier layer, which can effectively extract different features in the upper and lower eyelids and implement classification.

We employed a convolutional network model based on U-Net to segment the tarsus and meibomian glands. The normalized image was passed to an encoder transferred from ResNet34 to extract the feature, and then a segmentation decoder was employed, which could upscale the feature map to the same size as the original input image and implement the tarsus region. The skip connections also were used to promote the transmission of low-level features to higher levels. Another reconstruction decoder was cascaded after the feature extractor encoder to reconstruct the feature map to the original image so that we could help the encoder extract a more accurate feature map by the backpropagation of the error between the input image and the reconstructed image. More importantly, we imported the discriminator with fixed parameters from the pre-trained VAE-GAN as a quality inspector, embedding the shape constraint information into the neural network.

The tarsus region has a relatively definite shape. A VAE-GAN model was applied to extract and learn the shape feature. Then it could help the U-Net model to segment the tarsus region having a shape similar to the learned shape feature, as shown in Figure 3. In the VAE-GAN, the encoder and decoder formed a stacked autoencoder, which encodes tarsus images (Ground Truth, GT / Real) manually labeled by human experts into a feature space represented by a hidden vector Z, which is passed to the decoder for randomly generating eyelid tarsus images (Fake). Finally, the prior knowledge of the shape of the tarsus region is embedded in the convolutional neural network by training the discriminator of the real tarsus and fake tarsus. Here, the generator (G) and discriminator (D) formed a Generative Adversarial Network (GAN), and they increased the sensitivity and discrimination of the discriminator to the specific shape of the tarsus region in the process of fighting each other.

For training the U-Net, a specific loss function containing three items of Reconstruction Loss (RL), Dice coefficient Loss (DcL), and Discriminator Loss (DL) was applied, which was defined as:Loss=λ1∗RL(I,IRe)+λ2∗DcL(ISeg,IGt)+λ3∗DL(ISeg)
where I was the original image, the IRe was the reconstructed image from the feature maps, $I_{\mathrm{Seg}}$ was the segmented eyelid region, $I_{\mathrm{Gt}}$ was the real eyelid region (Ground Truth) labeled by human experts, and $\left[\lambda_{1},\lambda_{2},\lambda_{3}\right]$ was the weight vector.

The initial learning rate was 0.001, and we applied exponential decay to tune the learning rate dynamically. The training process stopped at around 100 epochs for tarsus area segmentation and 200 epochs for the meibomian gland to achieve the best performance. The research was performed with Anaconda Python3.6.5, PyTorch 1.6.0, and TorchVision 0.7.0 on a computer with Intel(R) Xeon(R) CPU E5-2650 v4 (Intel Corporation, Santa Clara, CA, USA), using NVIDIA Corporation GV100GL GPU (NVIDIA Corporation, Santa Clara, CA, USA) for training and testing.

### 2.4. Evaluation

The performance of the deep learning model was evaluated to demonstrate the similarity between the predictions and ground truth. Sensitivity, specificity, and accuracy were applied in evaluating the model. The image could be separated into the labeled area (L) and the rest area (R). True positive (TP) and true negative (TN) denoted the number of correctly classified pixels in the labeled or rest area. The number of pixels wrongly classified as labeled or rest area is false positive (FP) and false negative (FN).

We further calculated sensitivity ($\frac{TP}{TP+FN}$), specificity ($\frac{TN}{TN+FP}$), and accuracy ($\frac{TP+TN}{FP+TP+TN+FN}$). We also assessed the ability of the deep learning model in recognizing the tarsus area and meibomian gland by receiver operative characteristic (ROC) curves and the area under the curve (AUC).

## 3. Results

The first part of the model classifying upper and lower eyelids demonstrated high specificity, sensitivity, and accuracy of 100% on the test dataset. The results for upper and lower lid tarsus are demonstrated in Figure 4. The meibomian gland manual label and segmentation of the upper and lower lid meibography results are demonstrated in Figure 5. The results showed that the deep learning model could segment the upper and lower tarsus area and meibomian gland with high accuracy.

The ROC curves from the models segmenting the tarsus and meibomian gland areas are shown in Figure 6. The AUC values for models segmenting tarsus and meibomian gland areas were 0.985 and 0.938, respectively. The deep learning model achieved sensitivity and specificity of 0.975 and 0.99, respectively, with an accuracy of 0.985 and dice index of 0.94 for segmenting the tarsus area. For meibomian gland area segmentation, the model obtained a high specificity of 0.96, with high accuracy of 0.937, and Dice index of 0.94 and a moderate sensitivity of 0.751. In addition, the running speed of the model in this study can achieve 48 images per second.

## 4. Discussion

Recent research proposed a self-maintaining double vicious circle to manifest the onset and interacting pathophysiology of Meibomian gland dysfunction (MGD) and DED [3]. MGD arises from a combination of several mechanisms including eyelid and conjunctival inflammation, corneal damage, microbiological change, and tear film instability-associated DED. Meibomian gland changes serve as a crucial connecting point between two vicious circles [3]. The dropout of the meibomian glands leads to changes in the meibum causing tear film instability and imbalance. Thus, meibomian gland morphology evaluation is crucial in DED assessment. The present research trained a deep learning model to segment the tarsus and meibomian gland area automatically. The model demonstrated high accuracy, sensitivity, and specificity in recognizing the tarsus and meibomian gland area from meibography. With further improvement, the model could be applied to evaluate meibomian gland atrophy and loss in dry eye assessment.

Meibomian gland morphology evaluation is essential in DED assessment because the meibum is crucial in maintaining the tear film stability. Generally, the meibomian gland is visualized by infrared meibography, and then doctors manually grade the Meibomian gland loss. The manual grade induces inter- and intra-grader variability reducing the assessment accuracy. Previous studies have applied the traditional artificial intelligence method to automatically segment the meibomian gland and calculate the meibomian gland area as the percentage of the tarsus area [10,11,12]. However, these methods involve learning pre-defined features that require complicated pre-processing. Advanced research applied a deep learning model and trained the algorithm to automatically learn the feature and segment the meibomian gland from the meibography based on the label [17,18,19,20,21,22,23,24]. These studies applied different annotation methods. Several studies labeled the meibomian gland area as the whole district between the proximal ends of normal glands to the connecting line of gland orifices, or they marked the loss area between the gland ends to the edge of the proximal tarsal plate [17,20]. The area between each gland was considered. The atrophic meibomian gland could be thinner than the normal gland rather than shortening. Thus, merely considering the meibomian gland shortening would underestimate the gland atrophy. The gland-by-gland label could improve the accuracy of meibomian gland segmentation. Some studies annotate the region of individual meibomian glands using the polygon tool in Fiji (ImageJ) [18,23,24]. The semi-automated algorithm of the software might omit some small glands and less obvious parts of some glands. Thus, the area of the meibomian gland might be estimated inaccurately. The present research manually drew the contour of each meibomian gland as the ground truth to train the deep learning algorithm with a self-developed online coordination platform. Although laborious, manual annotation is expected to be more accurate. In the Hwang et al. study, researchers manually annotated each meibomian gland as precisely as possible with the Adobe photoshop brush and eraser tool, similar to our study [21].

Previous research achieved good performance by applying deep learning models to segment meibomian glands automatically. When annotating the whole region of the meibomian gland as the ground truth, Lin et al.’s study achieved an accuracy of 95.4% on meibomian gland atrophy segmentation [17], and the mAP (mean average precision) was 92% in Yang et al.’s study estimating meibomian gland loss [20]. In research annotating individual meibomian glands with ImageJ, Steven et al.’s [18] and Lin et al.’s [24] studies achieved a precision of 83% and 62.7%, respectively, to segment the meibomian gland area. Hwang et al. applied the manual annotation of each meibomian gland as the ground truth and achieved an accuracy of 67.7% [21], which is lower than our study. The present model segments the meibomian gland with outstanding performance, and the accuracy and specificity were 93.7% and 96%, respectively. The present research applied a VAE-GAN model to further augment the dataset. The encoder extracts real features from the dataset, and the decoder generates fake images based on these features. They are mixed and sent to the discriminator to improve the accuracy of identification during the adversarial iteration. However, the sensitivity was relatively low. We further analyzed the factors that may have affected the accuracy of segmentation during the processes. Firstly, much meibography has high reflective liquid points, which might be mistaken as meibomian gland hyper-reflection. Secondly, the contour of the meibomian gland was not apparent in some meibography, possibly due to image quality or capture problems. Thirdly, the appearance of eyelashes in the tarsus and meibomian gland areas could also interrupt the segmentation. Previous research applied a conditional generative adversarial neural network in MG segmentation [25]. In the GAN, the generator first segments the MG from the input, followed by a separate discriminator to determine each pixel accurately. The generator and discriminator are trained separately, which is different from our method.

With further improvement, the present automatic meibomian gland segmentation model could be profoundly applied in ophthalmologic research and clinical settings. Identification and assessment of meibomian gland loss are crucial in dry eye assessment and decision making during treatment. Traditionally, doctors manually grade the meibomian gland loss from meibography. The subjective grading causes variability between graders, and the discrete score system is less accurate and inconvenient for post-treatment evaluation. Previous research applied self-developed semi-automatic software to assess the meibomian gland following topical eye drops and eyelid warming treatment in patients with meibomian gland dysfunction [26,27]. The present algorithm could automatically segment tarsus and meibomian gland areas rapidly and accurately, facilitating the calculation of meibomian gland loss proportion. During the clinical trials, we could apply the software to rapidly enroll patients with meibomian gland loss of certain severity and precisely evaluate the changes in the meibomian gland following the treatment. The software could also be applied to predict the ocular surface circumstances in patients planning to receive ocular surface surgeries or wearing contact lenses. It is more convenient and sensitive for research purposes to apply a continuous indicator in meibomian gland evaluation for group comparison. MGD is also observed in patients with systemic diseases, including primary Sjögren’s Syndrome (pSS) [28]. Patients with pSS demonstrate reduced Meibomian gland secretion and gland dropout. The mechanism of MGD might be due to hormone changes and inflammatory cytokines release in patients with pSS [28]. Thus, the automatic Meibomian gland assessment tool could also be applied to pSS evaluation.

There are certain limitations in the present research and application of the method. Firstly, the model was only trained and tested with the meibography from a single machine. Thus, the generalization of our model has not been validated with a different external dataset. Future research would further include meibography from other centers and machines to improve the model. Secondly, calculating the meibomian gland area and its proportion to the tarsus area has not been associated with clinical symptoms. Further research should verify the normal value of the meibomian gland area and its physiological changes with age so that the software could better serve clinical and research usage. As for the method, the model could only quantitatively analyze the area of the meibomian gland. However, certain morphological abnormalities, including meibomian gland distortion, thinning, and dilatation, should be considered when analyzing the meibography [29]. We would further improve the algorithm in identifying abnormal morphological alternation of the meibomian gland to promote its efficiency in the clinical setting.

## 5. Conclusions

The present research trained a deep learning model to segment tarsus and meibomian gland areas from infrared meibography automatically, and the model demonstrated outstanding accuracy in segmentation. With further improvement, the model couldpotentially be applied to assess the meibomian gland that facilitates dry eye evaluation in various clinical and research scenarios.

## Figures and Tables

**Figure 1 jcm-12-01053-f001:**
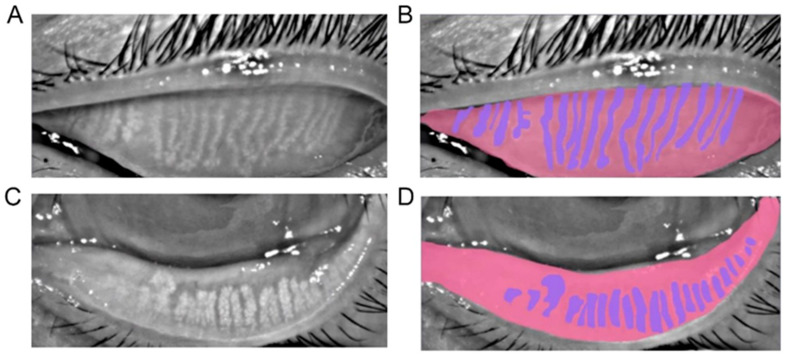
(**A**,**B**): A representative of meibography of the upper tarsus and its annotation; (**C**,**D**): A representative of meibography of the lower tarsus and its annotation.

**Figure 2 jcm-12-01053-f002:**
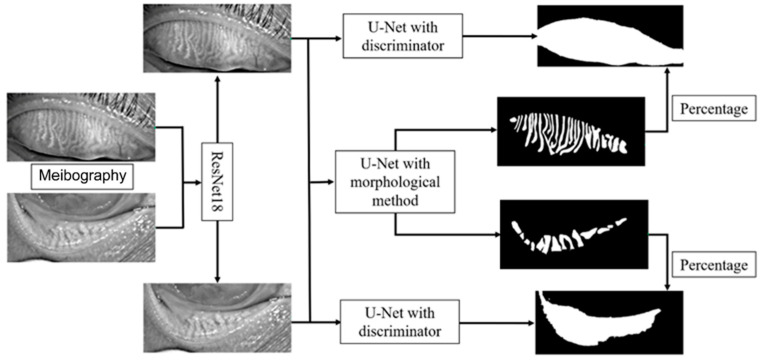
The framework of the deep learning model.

**Figure 3 jcm-12-01053-f003:**
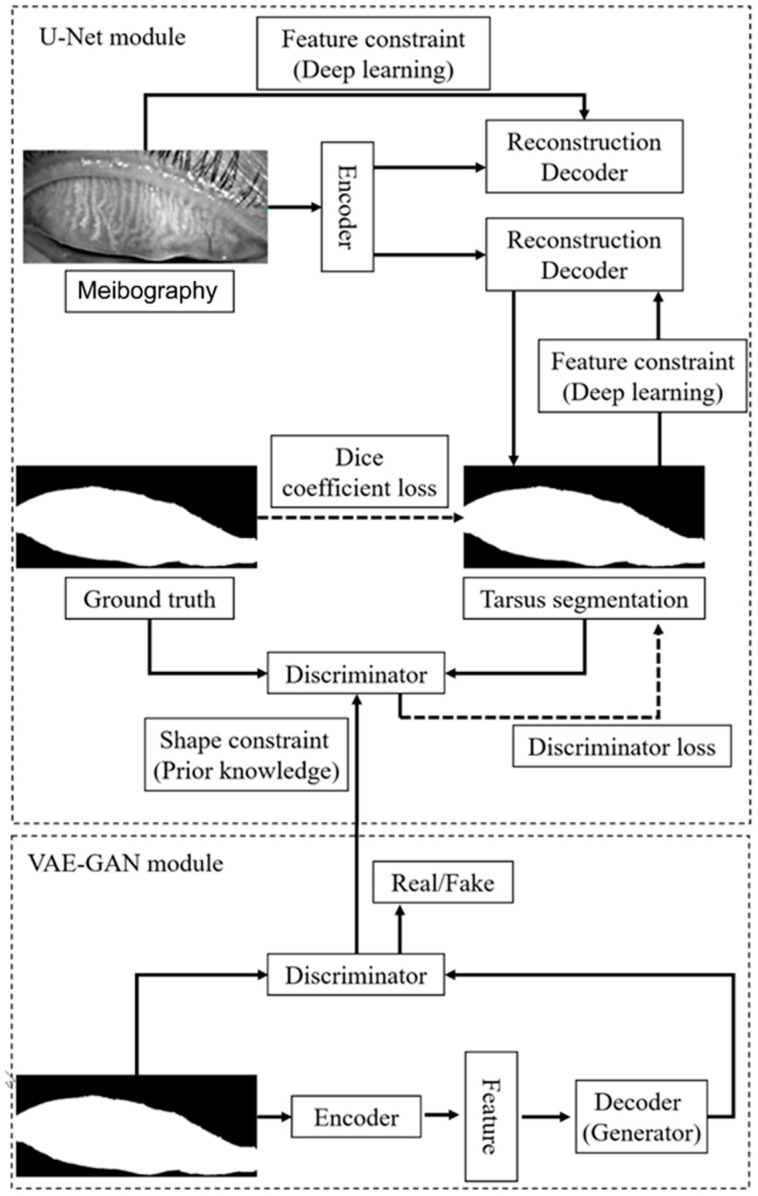
The framework of VAE-GAN and U-Net.

**Figure 4 jcm-12-01053-f004:**
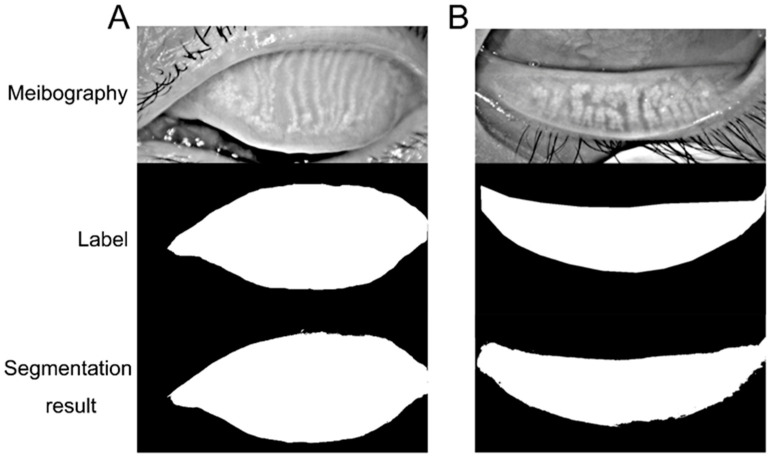
(**A**): The representative of meibography, label, and segmentation results for upper lid tarsus; (**B**): The representative of meibography, label, and segmentation results for lower lid tarsus.

**Figure 5 jcm-12-01053-f005:**
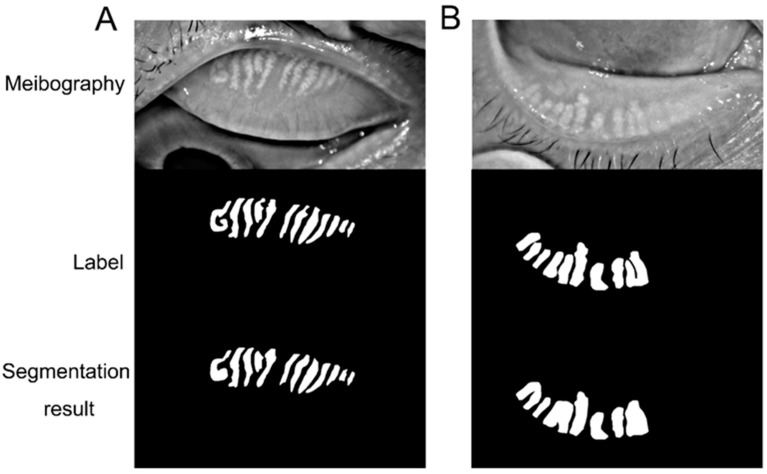
(**A**): The representatives of meibography, manual label, and segmentation results for upper and meibomian gland; (**B**): The representatives of meibography, manual label, and segmentation results for lower meibomian gland.

**Figure 6 jcm-12-01053-f006:**
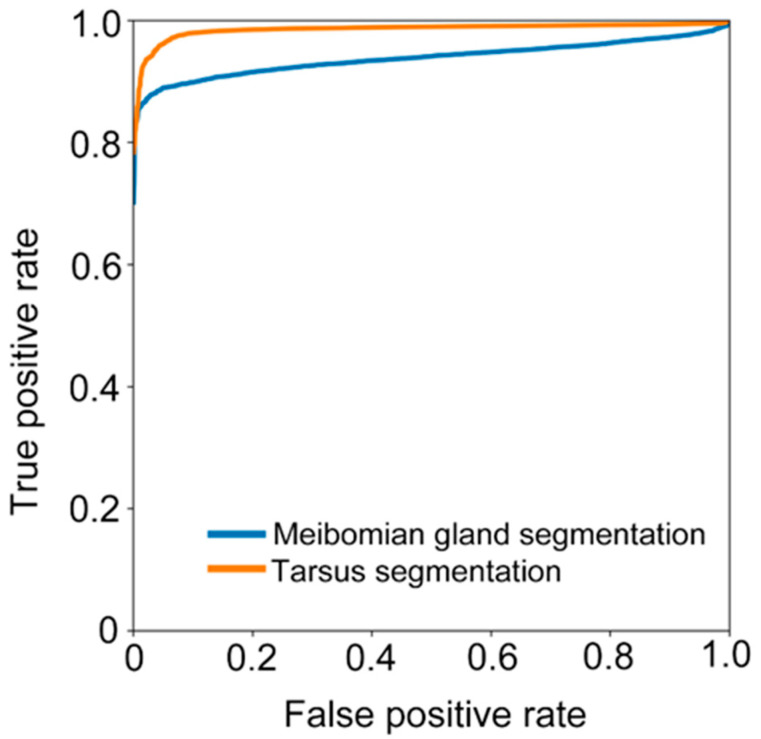
The receiver operative characteristic (ROC) curves for the models segmenting the tarsus and meibomian gland area.

## Data Availability

The analysis data used in this study are available from the corresponding author upon request.

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
