# Peer review of "A Deep Learning Model for Evaluating Meibomian Glands Morphology from Meibography"

_jcm, 2023, doi:10.3390/jcm12031053_

Round 1
Reviewer 1 Report
The authors have done an interesting study which could help towards the diagnosis of dry eye. commendable amount of work. However, there are queries which the authors can address.
1. How the current paper is different than the paper published by Setu et al., 2021 can be explained in detail in the introduction section (Setu, M.A.K.; Horstmann, J.; Schmidt, S.; Stern, M.E.; Steven, P. Deep learning-based automatic meibomian gland segmentation and morphology assessment in infrared meibography. Scientific reports 2021, 11, 7649, doi:10.1038/s41598-021-87314-8).
2. In section 2.3, the line number 101-104 is not relevant.
3. The legend of figure 1 (line 98), figure 4 line (184-185) and figure 5 line (185-187) is given twice.
4. The manual drawing of meibomian gland contradicts the automation aspect of the model described in the introduction section of the article.
5. In section 2.1, What were the meibography score distribution of the images included in the training, validation and testing dataset respectively?
6. In section 2.3, What was the size of dataset after augmentation?
7. Was the developed deep learning model evaluated by k-fold cross validation?
8. The ground truth images were drawn and labelled manually by experts, so how does the model account for the possibility of inter-expert variability?
9. The model to classify the upper and lower eyelid is demonstrating an accuracy of 100%, is this perfect accuracy occurring due to overfitting? What is the accuracy of this model on the testing and validation dataset?
10. Since the model developed in this study is specific to segment tarsus and meibomian gland areas from infrared meibography of patients diagnosed with dry eye disease, how effective would the developed model be in a clinical or research setup to segment the meibography images of healthy individuals?
Reviewer 2 Report
Great job on the design and methodology of the study.
Can the authors explain, how is it different from he work done by Deng et al https://doi.org/10.1016/j.eclinm.2021.101132 which looks at similar characteristics of meibomian glands and have actually comapred it in symptomatic and asymptomatic patients.
Round 2
Reviewer 1 Report
The manuscript can be accepted.